# Sex Chromosomes Are Severely Disrupted in Gastric Cancer Cell Lines

**DOI:** 10.3390/ijms21134598

**Published:** 2020-06-28

**Authors:** Sooeun Oh, Kyoungmi Min, Myungshin Kim, Suk Kyeong Lee

**Affiliations:** 1Department of Medical Life Sciences, Department of Biomedicine & Health Sciences, College of Medicine, The Catholic University of Korea, 222 Banpo-daero, Seocho-gu, Seoul 06591, Korea; dhtndms2@naver.com (S.O.); kyoungmimin@gmail.com (K.M.); 2Department of Laboratory Medicine, Seoul St. Mary’s Hospital, College of Medicine, Catholic University of Korea, Seoul 06591, Korea; microkim@catholic.ac.kr

**Keywords:** cell, sex difference, chromosome, loss of Y chromosome, gastric carcinoma, amelogenin, sex determining region Y, short tandem repeat, copy number variation, G-banding

## Abstract

Sex has not received enough attention as an important biological variable in basic research, even though the sex of cells often affects cell proliferation, differentiation, apoptosis, and response to stimulation. Knowing and considering the sex of cells used in basic research is essential as preclinical and clinical studies are planned based on basic research results. Cell lines derived from tumor have been widely used for proof-of-concept experiments. However, cell lines may have limitations in testing the effect of sex on cell level, as chromosomal abnormality is the single most characteristic feature of tumor. To examine the status of sex chromosomes in a cell line, 12 commercially available gastric carcinoma (GC) cell lines were analyzed using several different methods. Loss of Y chromosome (LOY) accompanied with X chromosome duplication was found in three (SNU-484, KATO III, and MKN-1) out of the six male-derived cell lines, while one cell line (SNU-638) showed at least partial deletion in the Y chromosome. Two (SNU-5 and MKN-28) out of six female-derived cell lines showed a loss of one X chromosome, while SNU-620 gained one extra copy of the X chromosome, resulting in an XXX karyotype. We found that simple polymerase chain reaction (PCR)-based sex determination gives a clue for LOY for male-derived cells, but it does not provide detailed information for the gain or loss of the X chromosome. Our results suggest that carefully examining the sex chromosome status of cell lines is necessary before using them to test the effect of sex on cell level.

## 1. Introduction

According to US Food and Drug Administration (FDA) reports, eight out of ten drugs are withdrawn from the market due to more severe side effects in women than in men. This is partially attributed to the fact that mainly male animals and male patients are used in the preclinical and clinical studies [1,2]. Evidence has accumulated showing that male-biased clinical studies result in unsuitable effects and/or unforeseen side effects for women. However, still there is much to improve in clinical research to close critical gaps on sex differences [3,4].

Among researchers using experimental animals, the notion that skewed results can arise from sex-biased study has been slow to spread [1,2]. Furthermore, sex has not been considered as an important biological variable among researchers using cells until fairly recently [5,6]. Most scientists neither design experiments to test the effect of cell sex nor report the sex of cells they used, even though the sex of cells affects cellular behaviors such as proliferation, differentiation, response to stress, and apoptosis [7,8,9]. Our previous study [10] showed that approximately 15.5% of human cell lines and 80–90% of animal cell lines were sold without sex identification. The majority of primary cells and stem cells were also provided without sex information. However, considering sex in basic research is important as preclinical and clinical studies are designed based on the results obtained using cells.

Amelogenin, existing on X (*AMELX*; p22.3) and Y (*AMELY*; p11.2) chromosomes as homologous sequences [11], has been broadly used for cell sex determination by polymerase chain reaction (PCR) [12]. It is easy to simultaneously amplify the different size fragments of amelogenin genes located on X and Y chromosomes (*AMELX* and *AMELY*, respectively) using only one pair of primers [13,14]. However, probing amelogenin alone can misidentify male-originated cells as female-originated ones when the cells have deletion in the Yp involving *AMELY* [15]. A well-known sex-determining gene, sex-determining region Y (*SRY*) located at Yp11.32 [16], has also been used to verify sex using the PCR method. As an alternative target for PCR-based sex determination, the homologue sequence of X-linked steroid sulfatase gene (*STS*) on Y chromosome was also reported [14]. These sex-chromosome-linked genes have been used for multiplex PCR amplification to determine the sex of objects [14,17]. Although the PCR approach is simple and cost-effective, designing a good primer pair is not easy for every purpose, and non-specific amplification can be a problem. In addition, as loss of Y chromosome (LOY) is often observed in male-derived cancer cell lines, it is difficult to define the sex of the cell through PCR experiments only.

The state of chromosomes has also been genetically investigated using various methods, such as fluorescence in situ hybridization (FISH) [18,19], chromosome banding techniques [20], copy number variation (CNV) analysis [21,22], and spectral karyotyping (SKY) [23,24,25]. Fluorescence in situ hybridization (FISH) analysis adopts probes complementarily hybridizing with specific genes on a chromosome. FISH efficiently detects chromosomal translocation, gene amplification, and deletion [18,19], but high background fluorescence and nonspecific signal detection are disadvantages. G-banding is a cytogenetic analysis that helps to determine the number and appearance of chromosomes to produce a visible karyotype [26]. Genome phenotype, such as duplication (gain) or deletion (loss), can be analyzed using copy number variation (CNV) analysis [21]. This method also has limitations of poor sensitivity and precision. Methods for detecting various X- or Y-linked genes have been also used to directly observe alteration in sex chromosomes in accordance with the purpose of the study [11,17,27]. As all these methods have strengths and weaknesses, it is desirable to use them in combination depending on the experimental purpose.

Here, we assessed the intactness of sex chromosomes of six male and six female-derived carcinoma cell lines which are commercially available and widely used for experiments. Five different methods were used to analyze the status of sex chromosomes of each cell line. In general, the results obtained using different analysis methods matched each other. Sex chromosomes in the cell lines showed various abnormalities, including Y chromosome loss.

## 2. Results

### 2.1. Polymerase Chain Reaction (PCR) of Sex Chromosomes

To check whether commonly used cell lines contain intact sex chromosomes, six male and six female gastric cancer (GC) cell lines were obtained for analysis. PCR amplification was performed for sex chromosome-linked genes previously employed for sex determination [23]: amelogenin (*AMELX* on the X chromosome; *AMELY* on the Y chromosome), sex-determining region Y (*SRY*), and the first intron site of steroid sulfate (*STS-1*) (Figure 1A). For *SRY* detection, two different sets of primer pairs were used as described in Methods section. Primer sets were designed so that the size of the PCR products differs depending on whether the gene is located in either the X chromosome or the Y chromosome: amelogenin_X-114 bp/Y-120 bp, *SRYa*_X-83 bp/Y-77 bp, *SRYb*_7th (as an internal control)-123 bp/Y-130 bp, and *STS-1*_X-158 bp/Y-166 bp [14].

The PCR products were analyzed by acrylamide gel electrophoresis (Figure 1B). The PCR products of all six female-originated cells were detected as a single band at the appropriate size. Two male cells, NCI-N87 and SNU-1, showed two PCR product bands as expected. However, the PCR products of the remaining four male cells (KATO III, SNU-484, SNU-638, and MKN-1) were detected as a single band each at the size where the amplification product from the X or the 7th chromosome is expected. In contrast, PCR amplicon from the Y chromosome was not detected in these cell lines. When the PCR product of *STS-1* gene for SNU-638 cell line was analyzed, an additional band amplified from the Y chromosome was also detected (Figure 1C).

### 2.2. Short Tandem Repeat (STR) Profiling of the Cell Lines

To make sure the PCR results of the four male cell lines (KATO III, SNU-484, SNU-638, and MKN-1) showing LOY were not due to cell line mix up, STR profiling was carried out for autosomal STR loci on chromosomes 2 (TPOX, D2S1338), 3 (D3S1358), 4 (FGA), 5 (D5S818), 7 (D7S820), 8 (D8S1179), 11 (TH01), 12 (vWA), 13 (D13S317), 15 (Penta E), 16 (D16S539), 18 (D18S51), 19 (D19S433), 21 (D21S11, Penta D), and sex chromosomes (amelogenin). In general, our STR results matched with the STR information the cell vendor provided for the 10 loci, including amelogenin (Table 1). In SNU-638, we detected the value 10, 13, and 14 at D5S818 STR, while only 10 and 14 were present in the STR information from the vendor. In addition, we could detect *AMELX* for SNU-484 and SNU-638, while amelogenin STR data were missing in the vendor’s information. Thus, *AMELX* but not *AMELY* was detected in all the four male-derived cell lines in consistent with our PCR results.

### 2.3. Fluorescence in Situ Hybridization (FISH) Analysis for Sex Chromosomes

To directly detect sex chromosomes of the three male-derived cell lines (SNU-484, SNU-638, and MKN-1), we performed FISH analysis using centromere evaluation probe X (CEPX) and Y (CEPY). Both the X and Y chromosome centromeres were detected by FISH in SNU-638. Interestingly, two instead of one X chromosome centromeres were observed in SNU-484 and MKN-1, while the Y chromosome centromere was undetectable (Figure 2).

### 2.4. Copy Number Variation (CNV) Analysis

Among the six male-derived cell lines, NCI-N87 and SNU-1 contained both the X and the Y chromosomes, matching with the sex of the cell donors. In SNU-638, the short arm seems to be lost from the Y chromosome judging by the log R ratio (LRR) value, which is lower than zero in this region. There was no detectable signal for the Y chromosome in SNU-484, KATO III, and MKN-1, implying LOY. One copy of the X chromosome was detected in NCI-N87, SNU-1, and SNU-638 (LRR = 0), while X chromosome gain may have happened in SNU-484, KATO III, and MKN1 (LRR > 0). Among the six female GC cells, partial X chromosome loss was indicated in SNU-5, MKN-28, and SNU-216 (LRR < 0). A female derived gastric carcinoma cell line, AGS, showed an LRR value of around zero and a value of 0.5 on the B allele frequency (BAF) plot, indicating a heterozygous genotype for the X chromosome in this cell line. For SNU-16 and SNU-620, hetero split on the BAF plot was observed together with near-zero LRR (Figure 3).

### 2.5. Giemsa Banding

Karyotyping was performed using chromosome banding techniques for SNU-484, SNU-638, SNU-5, MKN-28, SNU-216, and SNU-620. Giemsa banding revealed chromosomal abnormalities for both autosomes and the sex chromosomes in all the tested GC cell lines (Figure 4). The Y chromosome was undetectable in the male-derived SNU-484 and SNU-638 cell lines, while two X chromosomes were present in SNU-484 cells. There was only a single copy of the X chromosome in female-derived SNU-5 and MKN-28 cell lines (Figure 4). SNU-216 was observed to contain two X chromosomes, while there were three X chromosomes in SNU-620 consistent with CNV results.

## 3. Discussion

We tested the status of sex chromosomes in GC cell lines using several methods and compared the obtained results. In general, comparable results were obtained using five different analysis methods as described below. In the GC cell lines, gross chromosomal abnormality of sex chromosomes such as loss of Y chromosome (LOY) as well as loss or duplication of the X chromosome was observed in addition to small deletion and/or amplification in the sex chromosomes.

Our PCR, STR, and CNV analyses suggest that three (SNU-484, KATO III, and MKN-1) male cell lines suffer from LOY. FISH results showed that SNU-484 and MKN-1 cells do not contain the centromere of the Y chromosome, but they had two centromeres of the X chromosome, implying LOY and duplication of the X chromosome. G-banding assay confirmed that SNU-484 did not have the Y chromosome indeed, while two instead of one copy of the X chromosomes were present. LOY has been reported to be frequently observed in GC [28] as well as in other tumors. When sex chromosomes were analyzed by FISH in freshly isolated cells from gastric adenocarcinoma, six among eight GC cases showed LOY in contrast to normal gastric mucosa which had an intact Y chromosome [29]. The G-banding technique revealed three out of four primary gastric cancer samples from male patients had LOY [30].

At a glance, seemingly contradictory results were obtained from different assays we performed for a male cell line SNU-638. The G-banding result showed LOY for SNU-638, which is inconsistent with the report from the researchers who established this cell line [31]. In addition, our STR profiling results as well as STR information provided by the cell vendor indicate that *AMELY* is not present in SNU-638. However, the *STS-1* but neither the *AMELY* nor the *SRY* sequence from the Y chromosome was detected in PCR analysis for this cell line. FISH analysis also showed the presence of the centromere of the Y chromosome in SNU-638. The CNV results indicate that the short arm of the Y chromosome (Yp) where *AMELY* and *SRY* are located was lost, while the long arm of the Y chromosome (Yq) where *STS-1* is located was present in SNU-638. Collectively, these results imply that SNU-638 had lost Yp but maintained the centromere and Yq where the *STS-1* gene is located. Perhaps the remaining fragment of the Y chromosome was not detected by G-banding due to its small size.

Our PCR results for all the female cell lines show amplicons with appropriate sizes from the X chromosome. However, the PCR-based assay does not provide information about duplication or loss of the X chromosome. The G-banding results show that SNU-5 and MKN-28 had only one copy of the X chromosome even though they are derived from female GC patients. CNV analysis for these cell lines also supported loss of heterozygosity (LOH) with large deletion on the X chromosomes. For SNU-620, three copies of the X chromosomes were detected by G-banding, and the CNV assay indicated gain of the X chromosome.

The male cell lines (SNU-484, KATOIII, and MKN-1) with LOY seem to have duplicated the X chromosome judged from the CNV analysis and G-banding. Our results for KATO III are consistent with a previous report showing LOY and an XXX karyotype in this cell line [32]. Normally, one X chromosome of female cell undergoes inactivation (with about 15% of genes escape inactivation) for dosage compensation [33]. Although rarely studied in GC, it is suggested that gain of an extra X chromosome supports tumor progression in other cancers [34,35]. Recently, Xu et al. [36] suggested theoretical modeling of chromosomal evolution for free-living cell lines. Single nucleotide polymorphism (SNP) array analysis for 620 cell lines (279 from female and 341 from male tissues) showed that continually cultured cancer cells become “de-sexualized” through elimination of the Y and the inactive X chromosome. Following the de-sexualized state, many of the cells acquired a duplicate of their active X chromosome, supporting more efficient cell proliferation. Some of the cells we analyzed may also have undergone “de-sexualization” and then duplication of the active X chromosome. Further studies are warranted to find whether all of the X chromosomes within the cells containing the extra copy of X chromosome are active or not.

We found that simple PCR-based sex determination gives a clue for LOY for male-derived cells, but it cannot give detailed information for duplication or loss of the X chromosome. It seems that three (SNU-484, KATO III, and MKN-1) out of the six male-derived cell lines lost the Y chromosome completely, while one cell line (SNU-638) showed at least partial deletion in the Y chromosome. Meanwhile, two (SNU-5 and MKN-28) out of the six female-derived cell lines showed a complete loss of one X chromosome. One female-derived GC cell line SNU-620 had gained a copy of the X chromosome.

Our results suggest that cell lines may not be suitable to test the effect of sex on cell level as various abnormalities were found not only on the autosomes but also on the sex chromosomes. Primary cells are better choices than cell lines for sex-balanced cell experiments. However, it takes sizable blocks of time and efforts to isolate primary cells from tissues. The source of human primary cells is usually restricted, and obtaining pure primary cell population is not easy. In addition, primary cells have a finite expansion capacity and a short lifespan [37,38]. Thus, using primary cells in every experimental setting is limited in practice. When cell lines instead of primary cells are used to study sex differences, careful examination of sex chromosomes is strongly advised, using G-banding or whole genome sequencing.

## 4. Materials and Methods

### 4.1. Cells

Twelve gastric cancer cell lines (six were established from male patients: NCI-N87, SNU-484, SNU-1, SNU-638, KATOⅢ, MKN-1; six were established from female patients: SNU-5, MKN-28, SNU-216, SNU-16, AGS, SNU-620) were obtained from Korean Cell Line Bank (KCLB; Seoul, Korea). All of the cells were cultured in RPMI 1640 (Gibco BRL, Grand Island, NY, USA) supplemented with 10% fetal bovine serum (FBS) and antibiotics (100 U/mL penicillin and 100 g/mL streptomycin; Gibco BRL).

### 4.2. Genomic DNA

Cells were harvested, and genomic DNA was extracted using phenol:chloroform (Amresco, Solon, OH, USA) for the analysis of sex chromosomes.

### 4.3. PCR Detection of Sex Chromosomes

PCR was conducted to amplify genes showing size differences depend on whether they are located on the X or the Y chromosome:amelogenin, *SRY*, *STS-1*. Two primer sets were used to amplify different regions of the *SRY* gene—*SRYa* and *SRYb* [14]. The *SRYa* primer set amplified homologous sequences on the X or the Y chromosome, while the *SRYb* primer set amplified homologous sequences on the Y chromosome and the 7th chromosome. Amplified products of the 7th chromosome functioned as a positive control for successful amplification. A single PCR product band for each gene can be detected for DNA from a female-originated cell line as both of the X chromosomes have the same size genes. In contrast, PCR products having two different sizes can be observed for each gene if DNA from a male-originated cell line is used, as genes on the X and the Y chromosomes have different sizes.

The sequences of the primers were: amelogenin; 5′-GTTTCTTCCCTGGGCTCTGTAAAGAATAGTG-3′ and 5′-ATCAGAGCTTAAACTGGGAAGCTG-3′, *SRYa*; 5′-AGCCCCTGTGGAAAAAATTAGTTTT-3′ and 5′-AAAACACCATAGTTTTATATGGAGTAAAGGAAC-3′, *SRYb*; 5′-TTGTGCAGCCATCACCTCT-3′ and 5′-AAATCAGATTAATGGTTGCT-3′, *STS-1*; 5′-GTTTCTTCAGGAACATTTTGGAGCTAT-3′ and 5′-GGTTAGTTCAGGTGTAGACC-3′. Acrylamide gel was prepared using 40% acryl/bisacrylamide solution (Noblebio, Hwaseoung, Gyeonggi, Korea). PCR products were separated on a 15% acrylamide gel at 100 voltage. The bands in the acrylamide gel were detected by Gel Doc^TM^ (Bio-rad, Hercules, CA, USA) after staining with ethidium bromide (EtBr).

### 4.4. TR Profiling of the Cell Lines

We had Cosmogenetech Co. (Seoul, Korea) perform a STR profiling for the cell lines using the PowerPlex^®^ 18D System (Promega, Madison, WI, USA). The PowerPlex^®^ 18D System allowed co-amplification and four-color fluorescent detection of eighteen loci: TPOX, D3S1358, FGA, D5S818, CSF1PO, D7S820, TH01, vWA, D13S317, amelogenin, D16S539, D21S11, D18S51, D8S1179, D2S1338, D19S433, Penta D, and Penta E.

### 4.5. FISH Analysis for Sex Chromosomes

FISH analysis was performed with Centromere evaluation probe X/Y (CEPX/CEPY; Abbott Laboratories, IL, USA) in accordance with the manufacturer’s recommendations as previously reported [24]. Fluorescent images were captured with an Axio Imager 2 fluorescence microscope (Zeiss, Jena, Germany) and analyzed using Isis Fluorescence Imaging Platform (MetaSystems, https://metasystems-international.com/).

### 4.6. CNV Analysis

DNA samples were hybridized to Infinium Human Omni1M microarrays (Illumina, San Diego, CA, USA) in accordance with the manufacturer’s protocols. After whole-genome amplification, the product was fragmented, precipitated with 2-propanol, and resuspended in formamide-containing hybridization buffer. The DNA samples were denatured at 95 °C for 20 min, then placed in a humidified container for a minimum of 16 h at 48 °C allowing SNP loci to hybridize to the 50mer capture probes. For the single-base extension (Infinium II) assay, primers were extended with a polymerase and labeled nucleotide mix. The slides were then stained, washed with low salt wash buffer, immediately coated, and imaged on the Illumina iScan Reader. A single sample mode in which reference values are derived from canonical genotyping clusters (at 0, 0.5, and 1.0) created from clustering on ~120 normal reference samples was used. Individual copy number variation was detected using the software provided by the manufacturer (cnvPartition 3.2.0 plug-in of Illumina Genome Studio 2.0). Log R ratio and B-allele frequency were obtained from GenomeStudio. ASCAT (allele-specific copy number analysis of tumors) was then used in R to estimate aberrant cell fraction.

### 4.7. Giemsa Banding

Cultured cells were treated with 10 μg/μL colcemide (PAA Laboratories GmbH, Pasching, Austria) for 4 h. The cells were harvested and treated with hypotonic 0.075M KCl solution and fixed with Carnoy’s solution [11]. Banding cytogenetics was performed on G-banded metaphase chromosomes of the cells and karyotype was described according to the International System for Human Cytogenetic Nomenclature recommendations 2013 [20].

## Figures and Tables

**Figure 1 ijms-21-04598-f001:**
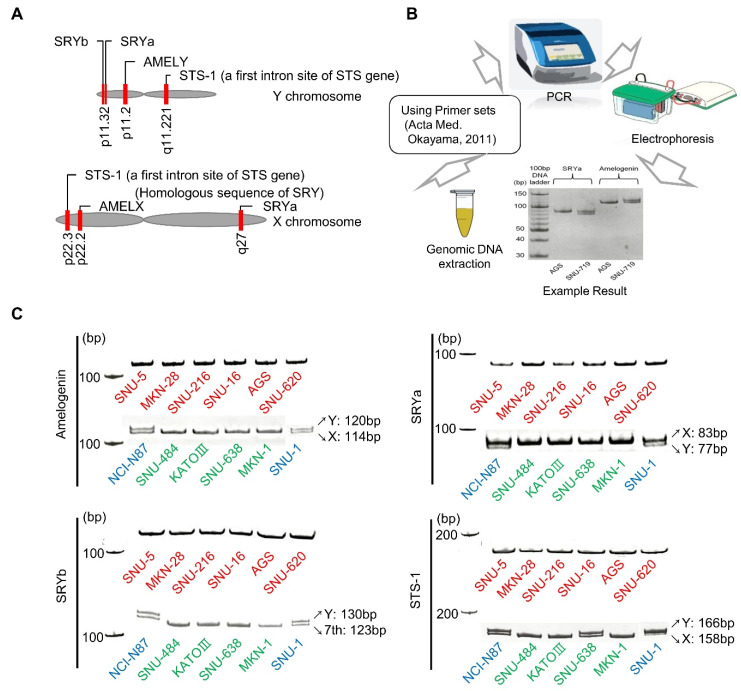
PCR analysis of sex-linked genes in GC cell lines. (**A**) Relative positions of the *SRYa*, *SRYb*, *AMELY*, and *STS-1* on the sex chromosomes. (**B**) Schematic diagram of the sex-linked gene detection using PCR–gel electrophoresis. (**C**) PCR products obtained using primers specific to amelogenin, *SRYa*, *SRYb*, and *STS-1* were separated on acrylamide gels. Size differences are noticeable between amplicons from the X and the Y chromosomes.

**Figure 2 ijms-21-04598-f002:**
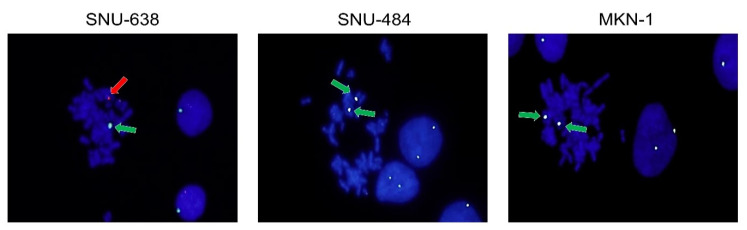
Fluorescence in situ hybridization (FISH) analysis of the X and the Y chromosomes. SNU-638, SNU-484, and MKN-1 cells were incubated with centromere evaluation probe X/Y to detect the X and the Y chromosomes. The X chromosome was stained green (indicated with green arrows) and the Y chromosome was stained red (indicated with a red arrow).

**Figure 3 ijms-21-04598-f003:**
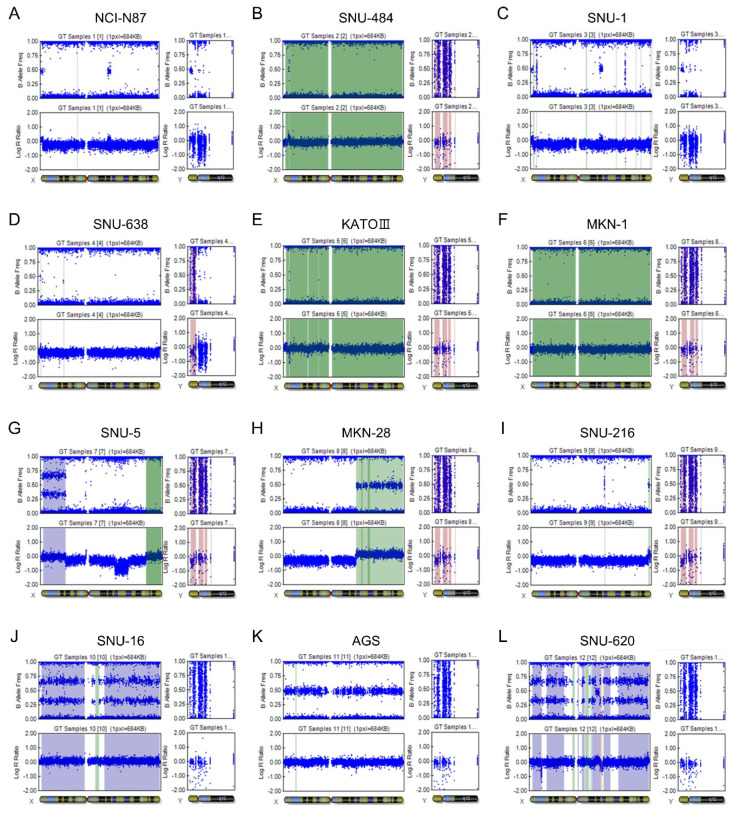
An illustration of B allele frequency (BAF; upper panel) and log R ratio (LRR; lower panel) values of the sex chromosomes in the 12 GC cell lines. A chromosome is displayed, from the short arm on the left to the long arm on the right. (Top plot) BAF values range from 0 to 1: Areas of homozygosity have BAF of 0 or 1; normal diploid regions have BAF of 0, 0.5, or 1; areas of allelic imbalance show intermediate values; homozygous deletions have no detectable signal so the calculated BAF appears as noise. (Bottom plot) LRR is the ratio of observed to expected intensities in log scale. An LRR of 0 represents the diploid state, whereas a value of >0 indicates a duplication has occurred and a value <0 designates a deletion has occurred. (**A**) NCI-N87, (**B**) SNU-484, (**C**) SNU-1, (**D**) SNU-638, (**E**) KATO III, (**F**) MKN-1, (**G**) SNU-5, (**H**) MKN-28, (**I**) SNU-216, (**J**) SNU-16, (**K**) AGS, (**L**) SNU-620.

**Figure 4 ijms-21-04598-f004:**
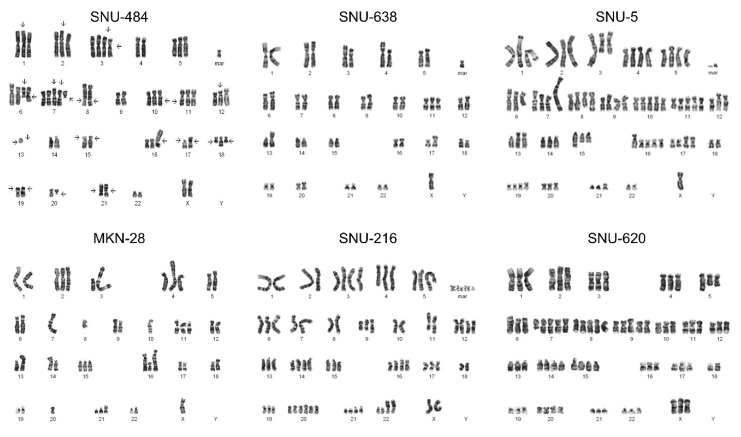
A complex karyotype of gastric carcinoma cell lines. Cells were treated with colcemid to hold the cells in the metaphase by inhibiting mitotic cycle. The cells were then Giemsa stained. Karyotype was described according to the International System for Human Cytogenetic Nomenclature recommendations. Arrows indicate abnormal chromosomes and possible breakpoints.

**Table 1 ijms-21-04598-t001:** Short tandem repeat (STR) profiling of four gastric carcinoma (GC) cell lines of male origin.

Locus	Cell Lines			
	KATOⅢ	SNU-484	SNU-638	MKN-1
TPOX	11	8	8, 11	8
D3S1358	15, 16	18	15	15, 17
FGA	23, 24	19, 23	17, 27	20, 23
D5S818	10, 11	10	10, 13, 14	11
CSF1PO	7, 11	9	10, 12	9, 12
D7S820	8, 12	12	9, 10	10
TH01	7, 9	7	7	9
vWA	14, 16	18	14, 19	16
D13S317	8, 12	11, 12	10	10, 12
**amelogenin**	**X**	**X**	**X**	**X**
D16S539	10, 12	13	11, 12, 13	11, 12
D21S11	30, 31	30	30.2, 31	29
D18S51	12	13, 15	14, 16	13, 16
D8S1179	13, 14	12, 14	11, 13	14
D2S1338	18, 20	23	18, 22	22
D19S433	13, 16	13	11, 14	13, 14
Penta D	13, 14	12	9	10
Penta E	13, 18, 19	22	14, 18	21

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
