# Peer review of "Sex Chromosomes Are Severely Disrupted in Gastric Cancer Cell Lines"

_ijms, 2020, doi:10.3390/ijms21134598_

Round 1

Reviewer 1 Report

The paper by Oh titled "Sex chromosomes are severely disrupted in gastric cancer cell lines" is a very interesting manuscript focused on the needs to test the effect of sex on cell only following the careful examination of sex chromosome status. Authors should report one of the first paper reporting a study on anomalies of Y chromosome in gastric cancer by castedo et al. (doi: 10.1016/0165-4608(92)90367-h). The manuscript is  accepted in the present form.

Author Response

  1. Authors should report one of the first paper reporting a study on anomalies of Y chromosome in gastric cancer by castedo et al. (doi: 10.1016/0165-4608(92)90367-h). The manuscript is accepted in the present form.

 → As advised, we cited the paper in our manuscript as shown in line 183.  

Reviewer 2 Report

GENERAL COMMENT

Evaluation of sex differences is a hot topic in medical research. The 2014 NIH announcement prompting researchers to assess sex differences in preclinical NIH-funded studies resulted in a major increase in the number of publications on sex differences aimed at identifying the influence of biological sex on study outcomes so preventing to reach erroneous conclusions.

On this background of evidence, Dr Oh and Colleagues from Korea examined the status of sex chromosomes in 12 commercially available gastric carcinoma cell lines using five  methods. Data have shown that comparable results were obtained using these different analysis methods. Simple PCR based sex determination gives a clue for LOY for male derived cells while not providing sufficient information regarding the X chromosome. The Authors conclude that marketed cell lines are unfit to test the effect of sex on cell level. The take-home message is that primary cells rather than cell lines should be used for sex balanced cell experiments.

 SPECIFIC COMMENT

I have found the following sentence unclear “ Our results suggest that cell lines may not [BE ?] suitable to test the effect of sex on cell level unless careful examination of sex chromosome status precedes [precedes what ?].

As evidences accumulate--> AS EVIDENCE ACCUMULATES

“ necessity of sex balanced clinical study design is accepted as a norm among clinical researchers nowadays” It is unfortunate that this IS NOT the case. Studies have shown. in the field of metabolism, that unaddressed research gaps still remain [Endocrinology. 2018;159(1):9-19. Hepatology. 2019;70(4):1457-1469]. Therefore the above unreferenced statement must be altered to be more consistent with this other Authors’ statement “Among researchers using experimental animals, the notion that skewed results can arise from sex biased study has been slow to spread”.

The conclusion is clear “primary cells rather than cell lines should be used for sex balanced cell experiments.”. Could these Authors explore this notion further by discussing what, based on their experience, opinion and available literature, prevents from this be put in practice ?

Author Response

  1. I have found the following sentence unclear “Our results suggest that cell lines may not [BE ?] suitable to test the effect of sex on cell level unless careful examination of sex chromosome status precedes [precedes what ?].

→ We revised the sentence as shown below.

“Our results suggest that carefully examining the sex chromosome status of cell lines is necessary before using them to test the effect of sex on cell level.” (line 31-33).

  1. As evidences accumulate--> AS EVIDENCE ACCUMULATES

→ As advised, we corrected the sentence (line 40-41).

  1. “necessity of sex balanced clinical study design is accepted as a norm among clinical researchers nowadays” It is unfortunate that this IS NOT the case. Studies have shown. in the field of metabolism, that unaddressed research gaps still remain [Endocrinology. 2018;159(1):9-19. Hepatology. 2019;70(4):1457-1469]. Therefore the above unreferenced statement must be altered to be more consistent with this other Authors’ statement “Among researchers using experimental animals, the notion that skewed results can arise from sex biased study has been slow to spread”.

→ We edited the manuscript and cited those papers as shown in line 42-43.

  1. The conclusion is clear “primary cells rather than cell lines should be used for sex balanced cell experiments.”. Could these Authors explore this notion further by discussing what, based on their experience, opinion and available literature, prevents from this be put in practice?

→ We expanded the discuss on that point as shown below.

“Primary cells are better choices than cell lines for sex balanced cell experiments. However, it takes sizable blocks of time and efforts to isolate primary cells from tissues. The source of human primary cells is usually restricted and obtaining pure primary cell population is not easy. In addition, primary cells have a finite expansion capacity and a short lifespan.” (lines 228-231).